# Effects of Saccharides Supplementation in the Extender of Cryopreserved Rooster (*Gallus domesticus*) Semen on the Fertility of Frozen/Thawed Spermatozoa

**DOI:** 10.3390/ani11010189

**Published:** 2021-01-14

**Authors:** Olga Stanishevskaya, Yulia Silyukova, Nikolai Pleshanov, Anton Kurochkin, Elena Fedorova, Zoya Fedorova, Oksana Perinek, Anna Prituzhalova, Inessa Meftakh

**Affiliations:** Branch of the L.K. Ernst Federal Science Center for Animal Husbandry, Russian Research Institute of Farm Animal Genetics and Breeding, 196625 Saint-Petersburg, Russia; olgastan@list.ru (O.S.); klaus-90@list.ru (N.P.); kurochkin.anton.66@gmail.com (A.K.); osot2005@yandex.ru (E.F.); zoya-fspb@yandex.ru (Z.F.); odormidonova@mail.ru (O.P.); aklevakina14@mail.ru (A.P.); meftax@ya.ru (I.M.)

**Keywords:** semen, cryopreservation, roosters, motility, fertility, trehalose, gene pool preservation

## Abstract

**Simple Summary:**

This study was devoted to the task of developing new balanced media for cryopreservation of rooster semen, which is of key importance for maintaining the functionality of spermatozoa after thawing. Trehalose in the medium increases the stability of various macromolecules under various physical influences and is capable of forming a stable vitreous matrix in cells during low-temperature stress, which maximally neutralizes the damage and metabolic activity in a dehydrated state. Also, the role of trehalose is to stabilize phospholipids during cooling and prevent disruption of the bilayer structure of cell membranes. The developed LCM-T10 and LCM-T20 media provided significantly higher rates of egg fertilization (82–86%) compared to the LCM (control) medium (79%, *p* < 0.05). The fertility of eggs on the 5th day from the last insemination in the LCM-T20 group had the best indicators of 100% versus 86% in the control, with 55% versus 20% in the control on the 10th day. When using test media, high results of egg fertilization were achieved and the functionality of frozen/thawed spermatozoa in the genital tract of the chicken was prolonged up to the 15th day. The results are intended to be used to preserve rare and endangered chicken breeds in vitro.

**Abstract:**

The aim of this study was to create balanced media for the cryopreservation of rooster semen in pellets to maintain the functional state of the sperm after thawing. Fructose was replaced by trehalose in experimental media in proportions of 10% (LCM-T10) and 20% (LCM-T20), while LCM was used as a control. After artificial insemination of the hens, the eggs were incubated (n = 400). To determine the functional safety of spermatozoa in the genital tract of hens after 5, 10, and 15 days from the last insemination, we used a method for assessing the interaction of sperm with the perivitelline membrane. Significantly higher rates of egg fertilization (82–86%) were obtained when using LCM-T10 and LCM-T20 compared to control (79%, *p* < 0.05). Egg fertility on the 5th day from the last insemination with the LCM-T20 diluent reached 100% versus 86% in the control; on the 10th day, the fertility rates were 55% versus 20%, respectively. The best results for fertility duration were obtained by freezing spermatozoa with LCM-T20 medium. The numbers of interaction points of spermatozoa with the perivitelline membrane were as follows: on the 5th day from the last insemination with LCM-T20—461.5 ± 11.5 holes/cm^2^ (LCM-control—13.7 ± 2.7 holes/cm^2^), *p* < 0.01; on the 10th day with LCM-T20—319.3 ± 12.9 holes/cm^2^ (LCM-control—14.9 ± 3.5 holes/cm^2^); and on the 15th day with LCM-T20—345.2 ± 11.1 holes/cm^2^ (LCM-control—0 holes/cm^2^). In conclusion, the use of trehalose in LCM diluent medium can increase the fertility of frozen/thawed sperm and the duration of their fertility in the genital tract of hens.

## 1. Introduction

The creation of cryobanks of semen is one way to preserve biodiversity in poultry [1]. A cryobank of rooster gametes is an insurance reserve that allows for overcoming the threat of extinction of breeds and populations and preventing the possible elimination of unique combinations of genes and their alleles during reproduction under natural mating conditions; it is a resource in collecting data to create a single database of genetic documentation and provides material for deep molecular genetic studies to examine the genetic basis of the productive, adaptive, and ethological traits of small chicken breeds [2]. In addition, cryopreservation of bird semen remains the preferred method today, as it is non-invasive. Conservation of rooster sperm can also play an important role in breeding, contributes to cost savings by reducing the number of roosters in live breeding, and provides greater flexibility in special breeding programs [3].

Much research has been performed to determine the best conditions for storing and freezing rooster semen in order to minimize damage to gametes and maintain their ability to fertilize. The publications on the development and improvement of methods and techniques for cryopreservation of rooster semen reflect the problems faced by researchers [3]—namely, a significant decrease in the fertilizing ability of frozen/thawed semen, caused by damage to the sperm membranes, DNA fragmentation, and a decrease in the total number of active sperm in the semen. However, some researchers have been able to achieve acceptable activity indicators of frozen or thawed spermatozoa and, with them, to increase semen fertility. Depending on the freezing methods used by different researchers, individual and pedigree characteristics of egg fertilization are in the range from 2 to 85% [4,5,6], and the average level of egg fertility when using frozen/thawed semen is usually less than 30% [7]. Silyukova et al. [8] showed that it is possible to achieve a stable level of frozen/thawed sperm fertility of up to 65%. One way to solve the problem of low fertility is to improve the cryoprotective environment for rooster semen with components that will prevent spermatozoon death and damage during freezing, storage, and thawing of the semen [9,10]. According to Thananurak (2019), a balanced cryopreservation diluent is the key to maintaining the functionality of sperm after thawing [11].

The determination of the composition of a new medium for the dilution of rooster sperm with the goal of cryopreservation should first be based on maintaining the osmotic balance and the balance of the sperm enzyme system [12]. Such a composition will not only determine the physiological content needed to maintain the vital activity of spermatozoa, including a normal pH level and an indicator of osmolarity of the medium, but also bear a simultaneous cryoprotective function. During cryopreservation, stabilization of cell membrane proteins and their organoids is also necessary. It is known that the stability of protein conformation can be ensured through the use of certain saccharides [13], which can be used to develop conceptually new approaches to creating cryoprotective media. To preserve its structure when exposed to negative temperatures, the protein must be embedded in a rigid glassy matrix [13]. Saccharose [11] and trehalose [14] have been used as such components, performing a cryoprotective function in mammalian semen media. Disaccharides, especially trehalose and sucrose, are widely known as effective bioprotective agents with exceptional ability to neutralize cold stress [15,16]. It is known that a significant amount of trehalose in bacterial cells counteracts a change in osmotic pressure on cell membranes when exposed to temperature shock [17]. According to published data, trehalose increases the stability of various macromolecules through predominant hydration during various physical influences [18] and is capable of forming a stable glassy matrix with extremely low molecular mobility during low-temperature stress, which maximally neutralizes the damage and metabolic activity in a dehydrated state [19]. The role of trehalose is also to stabilize phospholipids during cooling and to prevent disturbance of the bilayer structure of cell membranes [20,21,22].

Considering the physicochemical properties of trehalose, a number of researchers have tried to use it as a cryoprotectant of non-permeant action in a medium based on Lake diluent for the cryopreservation of rooster semen [23] and the semen of turkeys and cranes [24], but they did not achieve results that are significant for practical use. The frozen/thawed semen viability rates were 23.5–38.5% among roosters, 31.5% among turkeys, and 15.5% among partridges, while cranes had the highest frozen/thawed semen rate at 68.0%.

The purpose of this study was to determine the possibility of using trehalose as a cryoprotectant of non-permeant action as part of a medium for the cryopreservation of rooster sperm based on the Russian development of Leningrad Cryoprotective Medium (LCM) [25].

## 2. Materials and Methods

### 2.1. Conditions of Hen-Keeping and Rooster Semen Collection

The experiment was conducted on chickens of the Rhode Island Red breed (♂ n = 5, ♀ n = 45), aged 32–36 weeks, contained in individual battery cages (size = 45 × 60 × 60 cm, photoperiod = 14 L:10 D, temperature 18 °C). The experimental population was obtained according to the standards accepted in the Centre of Collective Usage “Genetic collection of rare and endangered chicken breeds” of the Russian Research Institute of Farm Animal Genetics and Breeding—Branch of the L.K. Ernst Federal Research Center for Animal Husbandry (RRIFAGB). Sperm was collected twice a week in 10 mL glass vials by using the abdominal massage method of Burrows and Quinn [26]. In field conditions, an individual assessment of the quality of the semen was carried out by observing macroscopic and primary microscopic indicators (x200, Mikromed MS-12, Saint-Petersburg, Russia, 2019). The sperm selection criteria were volume 0.2–1.2 mL; sperm concentration ≥3.1 billion/mL (Accuread Photometer, IMV Technologies, Bellshill, UK, 2019); motility: 80–85% (Mikromed MS-12, Saint-Petersburg, Russia, 2019); and agglutination of no more than 10%. Then, to eliminate individual differences between cocks, sperm samples were combined and divided into three aliquots according to the experimental plan (see details in the following sections). The study was conducted in accordance with the principles of bioethics in accordance with Art. 5 part 2 of the European Convention for the Protection of Vertebrate Animals for Experimental and Other Scientific Purposes (ETS 123 1986).

### 2.2. Composition of Media for the Cryopreservation of Rooster Semen

The components of the Leningrad Cryoprotective Medium (LCM) for semen freezing (Tselutin 2013) per 100 mL of distilled water were as follows: monosodium glutamate 1.92 g (114 mM), fructose 0.8 g (44 mM), potassium acetate 0.5 g (51 mM), polyvinylpyrrolidone 0.3 g (8.3 µM), and protamine sulfate 0.032 g (3.27 µM). For the experimental diluents LCM-T10 and LCM-T20, the composition was calculated with partial replacement of fructose with trehalose: fructose 0.72 g (40 mM) + trehalose 0.166 g (4.8 mM) and fructose 0.64 g (36 mM) + trehalose 0.326 g (9.5 mM), respectively; the other diluent components remained unchanged.

### 2.3. Semen Freezing and Thawing

The semen was frozen under production conditions. Diluted semen samples were equilibrated from 18 °C to 5 °C for 40 min. After cooling, dimethylacetamide (DMA, Sigma Aldrich, St. Louis, MO, USA) was added to each sample at a final concentration of 6%. After adding DMA, the samples were incubated at 5 °C for 1 min. Freezing was carried out in pellets by directly dripping the semen into liquid nitrogen. The starting position of the glass Pasteur pipette with semen was monitored using a digital manual temperature indicator with a sensor (THERM 2420, AHLBORN, Holzkirchen, Germany, 2019). In the region where the pipette was placed above the nitrogen surface, the temperature was between −15 °C and −20 °C. The average semen digging rate was ~1.4 pellets per second. The pellets were placed in marked glass vials and stored at −196 °C for 28 days. The pellets were thawed on a heated metal plate at 60 °C (self-developed equipment, RRIFAGB, Saint-Petersburg, Russia, 1989).

### 2.4. Motility Evaluation of Frozen/Thawed Semen

Sperm motion parameters were evaluated in frozen/thawed sperm diluted a second time in each extender using computer-assisted sperm analysis (Motic BA410E, China, 2019, negative contrast, x200; digital input system BASLER acA1300) and software (ArgusSoft, Saint-Petersburg, Russia, 2020). For motility analysis, sperm samples were diluted 1:40 to a concentration of approximately 35–40 million sperm/mL and loaded onto a warmed (38 °C) 10-μm Maklera^®^ camera (Slovenia, 2020). The progressive mobility parameter was used. Each sample was evaluated twice. Average values were taken into account.

### 2.5. Artificial Insemination

In the experiment we used virgin hens at the age of 32–36 weeks, n ♀ = 45, with 15 hens in each experimental group. To test the cocks’ sperm fertility, hens were inseminated intravaginally according to the following scheme: two days in a row with a single dose of insemination of 0.04–0.07 mL of frozen/thawed semen (insemination dose was at least 70–80 million progressively moving spermatozoa), and then every tree days (Table 1). The total number of insemination days was five, and the insemination time was from 14:00 to 16:00. Collecting eggs for incubation began a day the after the first insemination and was performed daily for nine days. Eggs were incubated for six days to assess the fertility of frozen/thawed semen (n = 400 eggs). Fertility evaluation of the eggs collected on the 15th, 20th, and 25th days of the experiment was carried out when the eggs were broken; fertility (% fertile/incubated eggs) was determined by blastoderm development after artificial insemination.

### 2.6. Lifespan Assessment of Frozen/Thawed Sperm in the Genital Tracts of Hens

In each experimental group, the dynamic of the functional state of frozen/thawed spermatozoa in the hens’ genital tracts was determined according to the Bakst method [27] by using the perivitelline membrane of the yolk of eggs 4, 5, 10, and 15 days after the last insemination.

The yolks were carefully separated from the egg whites. A ring of filter paper was placed on the yolk in the area of the blastodisc and then cut with curved medical scissors with the sharp end along the outer diameter of the ring. The separated membrane was removed using tweezers and washed several times in a 0.9% NaCl solution at 4–5 °C until the yolk material was completely removed, then transferred to a glass slide. For better visualization under microscopy, the material was stained according to the following protocol: 30 μL of a 10% alcohol formalin solution was applied uniformly on a perivitelline membrane using a micropipette. After 15–20 s, the formalin was poured off and ~30–40 μL of Schiff’s reagent was added to the glass slide in the same way using a micropipette with the tip wrapped in aluminum foil (to minimize light entering the Schiff reagent). After the perivitelline membrane acquired a purple hue (~30 s), the excess Schiff’s reagent was washed off with distilled water, and the material was then dried in air for 5 min before microscopic examination.

The areas inside the filter paper ring (the area of study was 1 cm^2^) were evaluated using an Axio Imager microscope (Carl Zeiss Microscopy GmbH, Jena, Germany, 2010) in a dark field at a magnification of x200. The number of interaction points (holes) of spermatozoa with the perivitelline membrane was counted.

### 2.7. Statistical Analysis

For statistical data processing, the software applications Excel 2013 (Microsoft, Washington, DC, USA, 2016) and Statistica 7.0 (StatSoft, Tulsa, OK, USA, 2016) were used. The data on the volume of semen and average sperm concentration corresponded to the normal Gaussian distribution. Data are presented as mean values ± standard deviation (±SE). The data of the eggs’ fertility rate and the data of the number of interaction points (holes) of spermatozoa with the perivitelline membrane were not normally distributed, and nonparametric analyses were thus used. The association between egg fertility rates and the kinds of media (LMC and LMC-T10; LMC and LMC-T20; LMC-T10 and LMC-T20) was assessed using the chi-square test and was considered significant at *p* < 0.05. The differences between the samples of interaction points (holes) of spermatozoa with the perivitelline membrane were evaluated using the Mann–Whitney *U* test and were considered significant at *p* < 0.01.

## 3. Results

The average volume of semen obtained from each rooster was 0.75 ± 0.2 mL; the average sperm concentration was 3.11 ± 0.78 billion/mL.

The level of progressive sperm motility of the native semen in all three groups did not significantly differ and averaged 85%. In frozen/thawed spermatozoa, the progressive motility also did not significantly differ and amounted to 48–50% in all groups (Table 2). The LCM-T10 and LCM-T20 media (Table 3) ensured egg fertilization at the level of 82–86%, while the level for the LCM-control was 79% (*p* < 0.05). At 6 days of incubation, all embryos corresponded to developmental stages 29–30 according to Hamburger and Hamilton [28], and no cases of early embryonic mortality were detected. 

Differences in the dynamics of egg fertilization were thus established, reflecting the functionality of sperm in hens’ genital tracts, depending on the medium used for cryopreservation. The best indicator was achieved in the LCM-T20 group—on the 15th day from the last insemination, 15% of eggs received were fertilized vs. 0% in the LCM and LCM-T10 groups.

An increase in the duration of the functionality of spermatozoa in hens’ genital tracts using the LCM-T20 diluent was confirmed using the Bakst method [27] (Figure 1): the numbers of interaction points of spermatozoa with eggs’ perivitelline membranes per 1 cm^2^ in the LCM-control group and experimental LCM-T10 and LCM-T20 groups on the fourth day from the last insemination were 57.1 pcs./cm^2^, 300.5 pcs./cm^2^, and 501.4 pcs./cm^2^, respectively. On the fifth day after insemination, in the LCM group, a decrease in the number of interaction points to 13.7 pcs./cm^2^ was noted. In the LCM-T10 group, the decrease was also significant, to level 139.2 pcs./cm^2^, and in the LCM-T20 group, the decrease reached a level of 461.5 pcs./cm^2^. On the 10th day, the dynamics continued. On the 15th day after the last insemination, an absence of interaction points was observed in the control group, and in the experimental LCM-T10 and LCM-T20 groups, the levels were 91.3 pcs./cm^2^ and 345.2 pcs./cm^2^, respectively. The differences between groups were significant at *p* < 0.01.

## 4. Discussion

Cryopreservation of rooster gametes can be carried out according to different protocols with different results. However, the problem of reduced functionality of sperm after a freeze/thaw cycle has not yet been resolved. The viability of embryos obtained from the use of cryopreserved sperm is low due to DNA fragmentation [29]. Improving the composition of diluents for cryopreservation, improving the selection of cryoprotectants, and modernizing freezing methods (in straws, pellets, and fast or slow protocols) [30,31] will allow further advancements in this field of research. The properties of trehalose make it possible to reduce the molecular mobility of the vitreous matrix, and this suggests that trehalose may be effective for maintaining the protein structure during prolonged storage at low temperatures [14,23,32].

In our study using trehalose in a medium for freezing rooster semen and using our protocol, high rates of progressive motility of frozen/thawed spermatozoa were obtained, at the level of 48–50%, but none of the proven media options (LCM-control, LCM-T10, and LCM-T20) had an effect on this parameter. Experiments using trehalose in a medium for freezing rooster semen were previously conducted by Mosca et al. [23,32] on roosters of a commercial cross (Lohmann), and positive motility preservation results were obtained and proved, confirming positive synergic action of the penetrating cryoprotective agent dimethylacetamide and the impenetrable cryoprotective agent trehalose on the quality of chicken sperm frozen/thawed into French straws. However, the progressive motility of frozen/thawed spermatozoa was at the level of 4.8–5.6%. It should be noted that in our study the concentrations of trehalose in the experimental media were 4.8 mM (LCM-T10) and 9.5 mM (LCM-T20), which are 10 or more times lower than that used in the experiment by Mosca et al. (100 mM) [23].

However, the preservation of frozen/thawed semen motility and progressive motility does not mean preservation of the spermatozoon integrity, chromatin integrity, or, as a result, functionality. Therefore, the main goal of our study was to determine the effect of trehalose in the composition of the medium for freezing rooster semen on the preservation of semen functionality, i.e., on fertility rates. The egg fertilization rates after artificial insemination of hens with frozen/thawed semen were 82–86% and were comparable to the average values from the use of native semen (fertilization rates 95–98%); further, the embryo development stage corresponded to their age.

We assume that the cryoprotective effect was achieved due to the presence of trehalose in the mitochondrial matrix of cells, since, according to Liu [33], trehalose provides increased stress resistance during freezing and dehydration of mitochondria. This suggests that the use of trehalose during cryopreservation of rooster gametes will preserve the integrity of the mitochondria of spermatozoa, too, which, in turn, will determine their energy status required to perform their motor function after the freeze/thaw cycle. A study by Zhang [34] showed that the use of trehalose encapsulated in thermosensitive nanocapsules to deliver a sufficient number of cellular organelles is likely to eliminate highly toxic intracellular cryoprotectants, which is promising for the preservation of reproductive cells for obtaining viable offspring.

Therefore, we consider it promising to continue research on the use of various combinations of saccharides in cryoprotective media for freezing rooster semen.

## 5. Conclusions

The presented results demonstrate the effectiveness of using trehalose disaccharide as a diluent for the cryopreservation of rooster semen. When using trehalose disaccharide at a concentration of 0.166 g (4.8 mM) in combination with fructose per 100 mL of distilled water in the diluent, the egg fertility was 82%; when using trehalose at a concentration of 0.326 g (9.5 mM) per 100 mL, egg fertility was highest—86% versus 79% in the control. 

The obtained high level of egg fertilization (82–86%) when using trehalose as part of the medium for cryopreservation of rooster semen, in combination with cryopreservation in pellets by directly dripping the semen into liquid nitrogen, provides grounds for the use of frozen/thawed rooster semen for breeding purposes in poultry. In our study, a significant increase in the duration of the functionality of frozen/thawed sperm in the genital tracts of hens was noted when using the LCM-T20 diluent composition (55% at the 10th day from the last insemination). Additional studies are required to determine the dynamic relationship between the fertilizing ability and the functional motility of frozen/thawed sperm in the genital tracts of hens when using experimental diluents. The technology being developed for the conservation of reproductive material in domestic bird species (Gallus gallus) can probably be successfully applied to the conservation of wild species (Galliformes) at risk of extinction.

## Figures and Tables

**Figure 1 animals-11-00189-f001:**
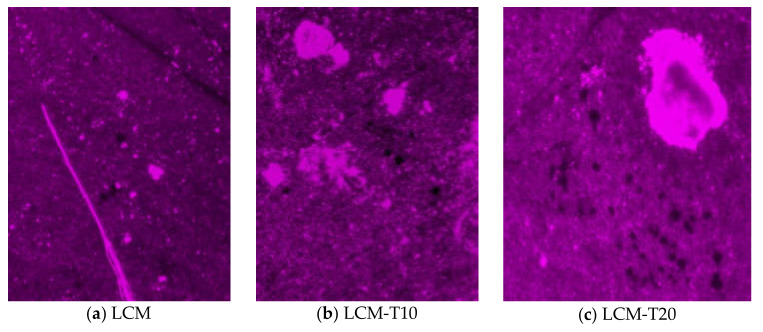
Images of the interaction points of the sperm with the vitelline membrane on the 15th day after insemination (a fragment of the visual field) depending on the cryopreservation medium used (LCM, LCM-T10, and LCM-T20 media).

**Table 1 animals-11-00189-t001:** Scheme of insemination of hens, collection of eggs for incubation, and collection of eggs to assess the survivability of spermatozoa in the hens’ genital tracts.

	Day of Experiment														
	1st	2nd	3rd	4th	5th	6th	7th	8th	9th	10th	11th	14th	15th	20th	25th
Insemination	+	+		+			+			+					
Collecting eggs for incubation			+	+	+	+	+	+	+	+	+				
Collecting eggs for blastodisc assessment												+	+	+	+

**Table 2 animals-11-00189-t002:** Indicators of the total motility of rooster semen (%) depending on the diluent used for cryopreservation.

Semen	Average Semen Motility %		
	LCM-Control *	LCM-T10 (Trehalose 10%)	LCM-T20 (Trehalose 20%)
Native	85	85	85
Frozen/thawed	48	48	50

* LCM-control, no additive; LCM-T10 extender supplemented with 4.8 mM trehalose; LCM-T20 extender supplemented with 9.5 mM trehalose.

**Table 3 animals-11-00189-t003:** Egg fertility rates (%) obtained by artificial insemination of hens with frozen/thawed rooster semen using different concentrations of trehalose in the diluent.

Medium	Eggs Laid, psc.	Egg Fertility Rates, %															
		Day of Experiment															Results for the Entire Period of the Experiment %
		1	2	3	4	5	6	7	8	9	10	11	14	15	20	25	
LCM-control *	130	x	x	73	100	100	90	86	91	100	100	100	90	86	20	0	79
LCM-T10	131	x	x	76	100	100	90	100	100	92	100	100	95	92	55	0	82
LCM-T20	139	x	x	67	100	92	100	100	100	100	100	100	100	100	50	15	86

* LCM-control, no additive; LCM-T10 extender supplemented with 4.8 mM trehalose; LCM-T20 extender supplemented with 9.5 mM trehalose.

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
