# Peer review of "Effects of Saccharides Supplementation in the Extender of Cryopreserved Rooster (Gallus domesticus) Semen on the Fertility of Frozen/Thawed Spermatozoa"

_animals, 2021, doi:10.3390/ani11010189_

Round 1

Reviewer 1 Report

This is an interesting manuscript related to the improvement of extenders for rooster semen cryopreservation with the use of threhalose. At general, manuscript is well written. Introduction is really informative and brings justificatives for the experimental design. There are, however, some points that should be revised before final decision.

Abstract seems a very unorganized. It should follows a chronologic description of methods and then, the results. Moreover, it lacks an objective conclusion highlighting the main findings.

Regarding methodology, please provide CASA settings and reference. Moreover, description of statistical analysis is so simplistic. Please inform if data were checked for normality and homocedasticity, inform if they were transformed, inform about the use of Variance analysis.

A discussion section is missing!

Conclusions should be more clear and objective. The need for additional studies will always be welcome, but this should be proposed at the discussions. In conclusions section, authors should highlight the main findings of the study and also point some direct applications for it.

Reviewer 2 Report

The manuscript by Stanishevskaya et al. reveals that the medium proposed by the authors (LCM-T20) containing 20% of trehalose is effective to obtain high levels of egg fertilization and it can be useful to sperm cryopreservation in domestic birds species. Although the results are relevant based on the fertilization rates showed, the authors have to rewrite the whole manuscript. English must be deeply reviewed by a native speaker. In the present form it is difficult to follow. I consider that the language used is not scientific in most of the cases, please check on other papers about cryopreservation of rooster semen how researchers refer to similar issues (for instance, papers from research group of Dr. Santiago-Moreno).

In additon, I consider that there are major corrections needed, which I detail below.

Summary:

The authors should highlight the importance of improving sperm cryopreservation process in avian species. The author did not mention the species, they used “male semen”. First part of the summary is general information and the authors should focus on their study and mainly in the novelty of this type of research in birds species. In addition, results should be showed in Abstract, nor in this section. Please avoid abbreviations such as “d” in line 19. Please, remind that the aim of this section is to show the relevance of the study to non expertise audience in the area, so I suggest to rewrite it.

Abstract:

This section should contain a briefly description of the aim, material and methods, results and conclusion. The authors only described the results. Please, rewrite this section.

Introduction:

Line 66: please change “reproductive cells” by “gametes”.

Line 68: please avoid “recent publications” when you provide a reference from 2016.

The authors should avoid expression such as “from our point of view” (line 52) or “In our understanding” (line 81) when they refered statement which are known that exists so they should provide reference to support them.

This section should be reduced, for instance, first paragraph could be removed since it does not provide any relevant information for the present study.

In general, but also in this section, I miss that the author highlight the species which is the aim of their research. In the second paragraph, again, they did not mention it till line 61.

Line 108: please change “exocellular” by “non-permeant”.

Material and Methods:

Please , remove the white page before Statistical analysis section.

Results:

Please, define semen activity, Do the author mean progressive motility??

Figures:

I do not understand the meaning of Figure 1. What indicate the yellow arrows? I have problem to follow the results with this figure.

Discussion:

There is no Discussion section. If the authors prefer to present the Result and Discussion sections together, this should be indicated. However, in both cases, the results are not properly discussed. The authors must compare their results with other previously reported. There are only few lines which I can consider as a Discussion, but in my opinion it is not enough.

Information provides in lines 219-230 is included in the Introduction, so please remove them.

Round 2

Reviewer 2 Report

The manuscript has been improved, however, I consider that there still are some issues that the authors should ammend.

Abstract:

Lines 27-29: please rewrite this sentence.

Lines 35-38: please rewrite this sentence.

Results:

Authors should remove the discussion of their results in this section and add this information to the Discussion one. Now, in the revised versión there is a Discussion section, thus, the Results should only show a description of the obtained results. Please rewrite carefully this section.

Discussion:

Please, see the previous comment and rewrite carefully Discussion section.

Author Response

Пожалуйста, посмотрите приложение

Round 3

Reviewer 2 Report

Although the authors have revised the paper satisfactorily, I still have some comments. Please see below. 

Line 28-29: please change the expression "controlled by...".

Discussion: the authors should present their results  at the beginining of the second paragraph and then discuss them. Otherwise it seems an Introduction instead of Discussion. 

Conclusions: this section should be reduced. 
